# Contrasting responses to aridity by different-sized decomposers cause similar decomposition rates across a precipitation gradient

Viraj R Torsekar[1,2]*[†], Nevo Sagi[1]*[†], J Alfred Daniel[3], Yael Hawlena[1], Efrat Gavish-Regev[3], Dror Hawlena[1,3]

[1]Risk-Management Ecology Lab, Department of Ecology, Evolution & Behavior The Alexander Silberman Institute of Life Sciences, The Hebrew University of Jerusalem, Jerusalem, Israel; [2]Theoretical Ecology and Evolution Lab, Centre for Ecological Sciences, Indian Institute of Science, Bengaluru, India; [3]The National Natural History Collections, The Hebrew University of Jerusalem, Jerusalem, Israel

*For correspondence:
viraj.torsekar@gmail.com (VRT);
nevo.sagi@mail.huji.ac.il (NS)

[†]These authors contributed equally to this work

Competing interest: The authors declare that no competing interests exist.

**Abstract** Litter decomposition is expected to be positively associated with precipitation despite evidence that decomposers of varying sizes have different moisture dependencies. We hypothesized that higher tolerance of macro-decomposers to aridity may counterbalance the effect of smaller decomposers, leading to similar decomposition rates across climatic gradients. We tested this hypothesis by placing plant litter baskets of different mesh sizes in seven sites along a sharp precipitation gradient, and by characterizing the macro-decomposer assemblages using pitfall trapping. We found that decomposers responded differently to precipitation levels based on their size. Microbial decomposition increased with precipitation in the winter while macro-decomposition peaked in arid sites during the summer. This led to similar overall decomposition rates across the gradient except in hyper-arid sites. Macro-decomposer richness, abundance, and biomass peaked in arid environments. Our findings highlight the importance of macro-decomposition in arid-lands, possibly resolving the dryland decomposition conundrum, and emphasizing the need to contemplate decomposer size when investigating zoogeochemical processes.

## eLife Assessment

This **fundamental** study substantially advances our understanding of the role of different-sized soil invertebrates in shaping the rates of leaf litter decomposition, using an experiment across seasons along an aridity gradient. The authors provide **compelling** evidence that the summed effects of all invertebrates (with large-sized invertebrates being more active in summer and small-sized invertebrates in winter) on decomposition rates result in similar levels of leaf litter decomposition across seasons. The work will be of broad relevance to ecosystem ecologists interested in soil food webs, and researchers interested in modeling carbon cycles to understand global warming.

## Introduction

Litter decomposition is a key process determining elemental cycling in terrestrial ecosystems (*Schlesinger and Bernhardt, 2020*). Decomposition is controlled by climate, litter quality, and origin, and the identity and abundance of microbial and faunal decomposers (*Bradford et al., 2016*; *Joly et al., 2023*; *Swift et al., 1979*). Climate regulates decomposition rates directly, but also indirectly

**eLife digest** In most ecosystems on land, it is largely small organisms such as microbes that break down dead plant material (known as plant litter) into nutrients that are recycled into the soil. Given that microbes need moisture to survive, scientists have long questioned how plant litter undergoes this decomposition in dry ecosystems.

Previous research focused primarily on how solar radiation and other environmental factors affect how quickly plant litter decomposes in these harsh conditions. However, another possibility is that larger decomposers, such as animals like beetles and termites that feed on dead plant material, are better adapted to arid conditions and may be more abundant in areas with low rainfall. As a result, plant litter in dry environments may decompose at similar rates to areas with higher rainfall.

Torsekar, Sagi et al. tested this idea by monitoring how quickly plant litter decomposed at seven sites with similar average temperatures but different rainfall levels. Dozens of baskets with different sized mesh – which excluded some or all animal decomposers based on size – were placed at each site and a technique called pitfall trapping was used to identify the decomposers at each site.

The experiments showed that plant litter broke down at similar rates across five of the seven sites, but decomposition was slower at extremely dry sites. In the winter, when rainfall is typically higher than at other times in the year, microbe decomposers played a bigger role in breaking down the leaf litter than in the (drier) summer months. On the other hand, animal decomposers were more abundant at sites with low rainfall than sites with higher rainfall. Furthermore, decomposition by animals at these arid sites during summer was just as fast as microbial decomposition at the wetter sites in winter.

The findings of Torsekar, Sagi et al. suggest that larger, animal decomposers compensate for the lower microbial decomposition of plant matter in ecosystems with little rainfall. In the future, a better understanding of how nutrients are recycled in dry areas will help predict how different ecosystems will respond to climate change.

by influencing food-web structure and dynamics (*Wu et al., 2021*). Thus, understanding how climate and decomposers interact is a key step in explaining variation in plant litter decomposition across ecosystems and seasons, and in forecasting the consequences of climate change and biodiversity loss on elemental cycling.

Theory suggests that decomposition is positively associated with moisture and temperature (*Swift et al., 1979*). Cross-site studies, reviews, and meta-analyses verified this global pattern, showing that plant litter decomposition in microbial litter bags is indeed faster under warm and wet conditions than under cool and dry conditions (*Berg et al., 1993*; *Meentemeyer, 1978*; *Bradford et al., 2017*; *Parton et al., 2007*; *Aerts, 1997*; *Zhang et al., 2008*). This well accepted realization implicitly assumes that microorganisms dominate plant litter decomposition, largely ignoring the growing recognition that animals may play an important role in litter cycling. This role includes mineralizing and excreting assimilated plant nutrients, fragmenting and partly decomposing plant material, and transporting detritus to microbial havens (*David, 2014*; *Frouz, 2018*; *Griffiths et al., 2021*; *Auclerc et al., 2022*; *Sagi et al., 2021*; *Coulis et al., 2016*; *Joly et al., 2018*; *Joly et al., 2020*).

Attempts to explore how climate affects faunal decomposition revealed a similar positive association with temperature and precipitation (*García-Palacios et al., 2013*; *Xu et al., 2020*). This global pattern, however, may be confounded by compiling the effect of all decomposer fauna together, ignoring the well-established understanding that soil animals of various size groups respond differently to climate (*Swift et al., 1979*; *Johnston and Sibly, 2020*). Specifically, larger arthropods can survive and remain active during hot and dry periods when smaller organisms cannot (*Cloudsley-Thompson, 1975*). Indeed, handful of evidence shows that macro-detritivorous arthropods dominate litter and wood decomposition in warm drylands, especially during warm and dry seasons (*Veldhuis et al., 2017*; *Sagi et al., 2019*; *Zanne et al., 2022*). This suggests that the conceptualization of how animals and climate interact to regulate decomposition rates requires considering the effects of meso-decomposers and those of macro-decomposers separately, particularly in warm drylands.

Detritivorous animals are expected to be exceptionally abundant in arid ecosystems where plant detritus is prevalent year-round but green plant material is available predominantly in short pulses

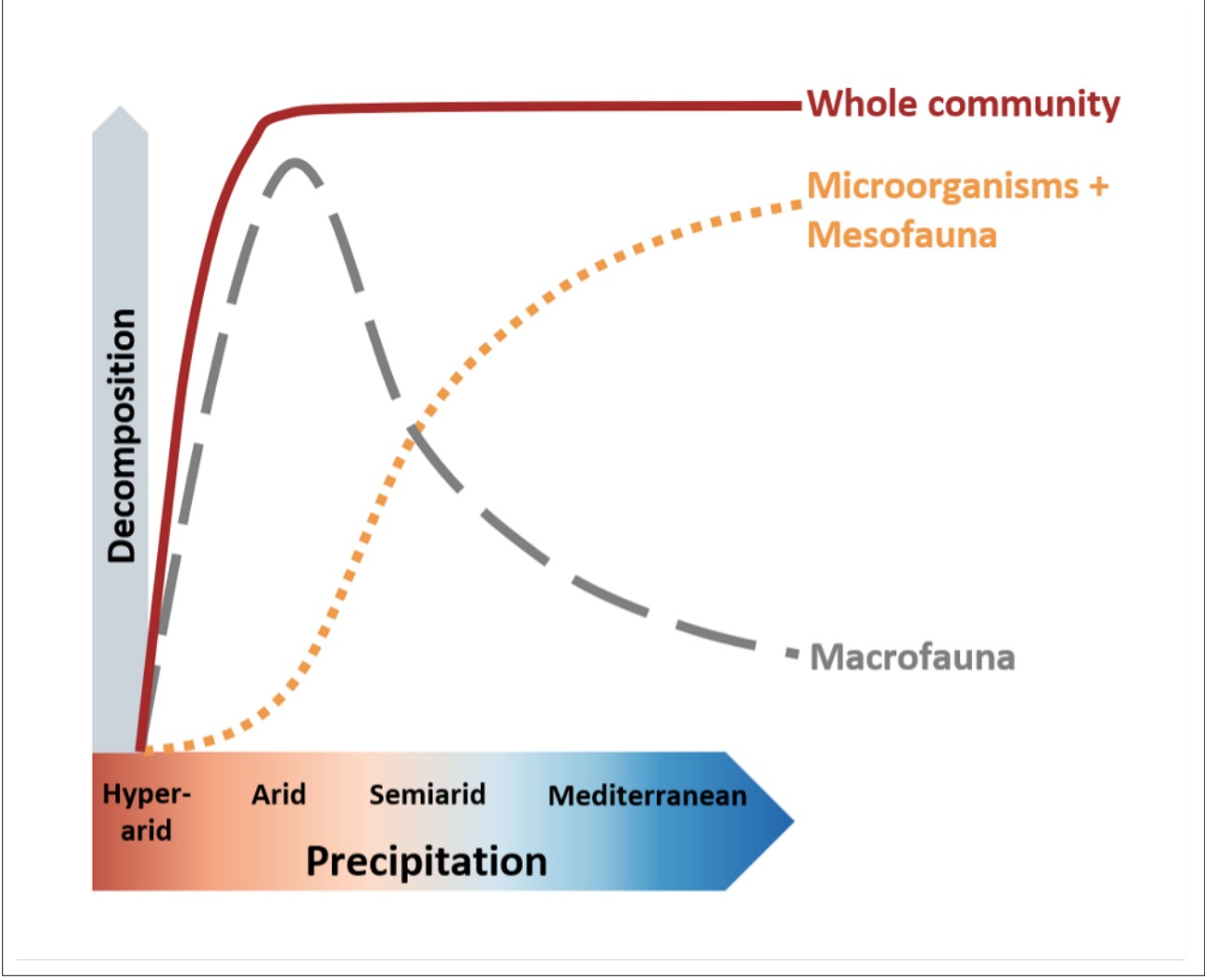

**Figure 1.** Hypothetical climate dependence of litter decomposition by microorganisms and mesofauna (dotted orange curve), by macrofauna (gray – dash), and by the whole decomposer community (maroon – solid).

following precipitation events (**Ayal et al., 2005**). Macrofauna are physiologically and morphologically more adapted to aridity than mesofauna (**Cloudsley-Thompson, 1975**). Moreover, their large size enables them to remain active during long dry periods by shuttling between existing and self-engineered climatic havens and the hostile foraging grounds on the surface (**Sagi and Hawlena, 2021**). Consequently, macro-decomposition should be especially important in hot moisture-deprived habitats and periods, whereas the activity of microbes and mesofauna is expected to be minimal.

The predicted negative association between moisture and macro-decomposition in drylands may be reversed in hyper-arid environments. In these environments, the extreme climatic conditions and scarce and unpredictable plant litter availability may limit macro-decomposer populations, diminishing macro-decomposition rates with increasing aridity. Consequently, and in sharp contrast to smaller organisms, macro-decomposition should follow a hump-shaped response to precipitation that peaks in arid ecosystems (**Figure 1**).

To test this novel hypothesis, we examined the climate dependency of plant litter decomposition by microorganisms, meso-decomposers, and macro-decomposers along a sharp aridity gradient spanning from mean annual precipitation (MAP) of 22–526 mm. This gradient represents hyper-arid, arid, semiarid, and dry sub-humid Mediterranean climates (**Table 1**, **Figure 2A**). We repeated the experiment during hot summer with no precipitation and again during cooler and wetter winter. We hypothesized that both microbial and mesofaunal decomposition should increase with increasing precipitation during the winter, but during the dry summer contribute only minimally to plant litter decomposition

**Table 1.** Properties of the seven experimental sites.

| Site | Abb. | Coordinates | MAT* [°C] | MAP* [mm] | AI$_U$* (MAP/PET) | Climate | Winter experiment | Summer experiment |
|---|---|---|---|---|---|---|---|---|
| Ramat Hanadiv | RH | 32°33'22.4"N 34°56'26.6"E | 20.2 | 526 | 0.518 | Dry sub-humid Mediterranean | 3.12.2020– 27.6.2021 | 27.6–27.10.2021 |
| Bet Guvrin | BG | 31°35'54.7"N 34°54'14.2"E | 20.9 | 403 | 0.370 | Semiarid | 2.12.2020– 13.6.2021 | 13.6–21.10.2021 |
| Havat Shikmim | HS | 31°30'49.7"N 34°41'18.8"E | 19.8 | 367 | 0.364 | Semiarid | 2.12.2020– 13.6.2021 | 13.6–21.10.2021 |
| Sayeret Shaked | SS | 31°16'05.7"N 34°39'12.9"E | 20.0 | 148 | 0.145 | Arid | 26.11.2020– 23.5.2021 | 23.5–21.10.2021 |
| Avdat | AV | 30°47'02.3"N 34°46'13.3"E | 18.7 | 84 | 0.089 | Arid | 26.11.2020– 23.5.2021 | 23.5–21.10.2021 |
| Meishar | MS | 30°27'04.2"N 34°56'03.0"E | 20.8 | 33 | 0.029 | Hyper-arid | 10.12.2020– 12.7.2021 | 12.7–8.11.2021 |
| Nahal Shita | NS | 30°08'29.4"N 35°07'36.6"E | 22.3 | 22 | 0.017 | Hyper-arid | 10.12.2020– 12.7.2021 | 12.7–8.11.2021 |

*Climatic data extracted from http://www.meteo-tech.co.il/hanadiv_new/hanadiv_en.asp (RH), courtesy of Shaily Dor-Haim (SS), and extracted from https://ims.gov.il/en (all other sites).

across the aridity gradient. In the dry summer, macrofaunal decomposition should follow a hump-shaped response to precipitation, increasing from hyper-arid to arid sites and decreasing gradually in more mesic semiarid and Mediterranean sites. We also predicted that the opposing climatic dependencies of macrofauna and microorganisms and mesofauna should lead to similar overall decomposition rates across the precipitation gradient except in the hyper-arid sites in which decomposers

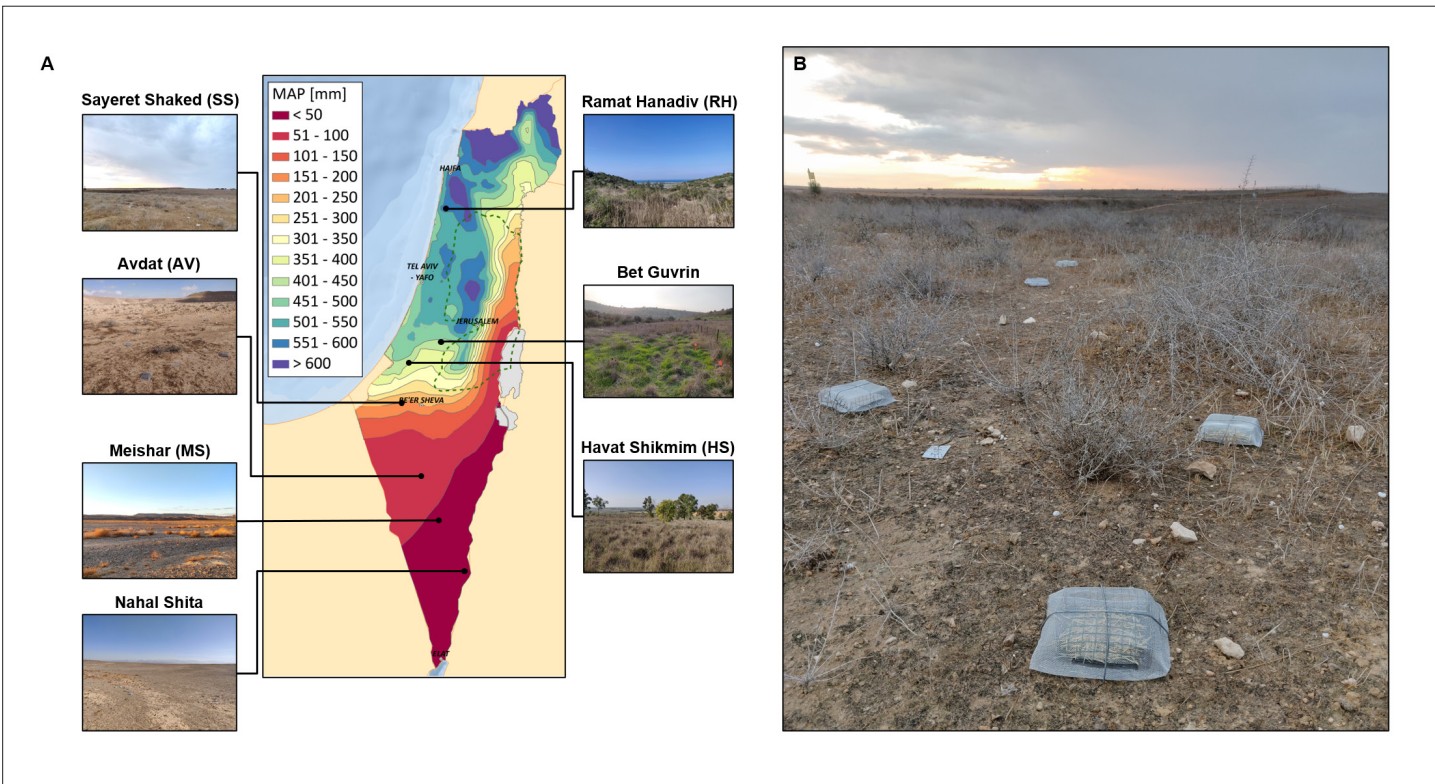

**Figure 2.** Illustration of the experimental design. (**A**) Locations and landscapes of the seven experimental sites across a precipitation gradient from 22 to 526 mm mean annual precipitation (MAP). (**B**) A block of three litter baskets in the Sayeret Shaked site. Macro-basket in front, meso-basket on the right, and micro-basket on the left. The precipitation map courtesy of the Hebrew University GIS center.

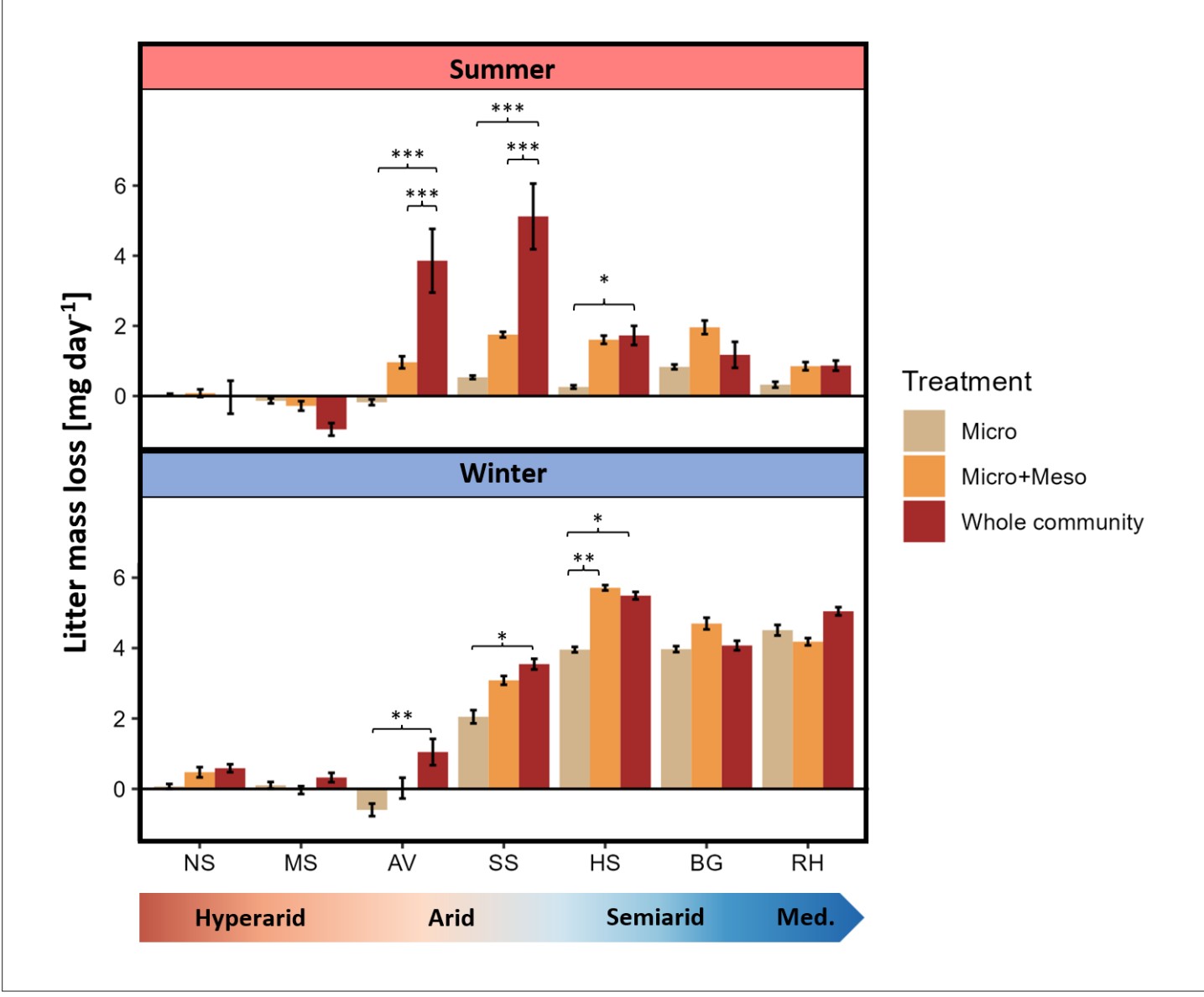

**Figure 3.** Litter removal rate (mean ± standard error [se]) from baskets with different mesh sizes across sites and seasons. Asterisks represent significant differences between mesh sizes within site and season: *p-value <0.05, **p-value <0.01, ***p-value <0.001 (Tukey's Honestly Significant Difference). Each bar represents 25 samples, total n = 1050. NS – Nahal Shita; MS – Meishar; AV – Avdat; SS – Sayeret Shaked; HS – Havat Shikmim; BG – Bet Guvrin; RH – Ramat Hanadiv. Negative values may represent cases in which physical cleaning and ash correction failed to correct for all dust accumulation on the litter or cases in which exogenous litter may have penetrated the baskets. Our findings were not sensitive to these negative values.

activity is predicted to be minimal regardless of organism size (*Figure 1*). To reveal the mechanism, we sampled macro-decomposers across the aridity gradient and the two seasons, using pitfall traps. We predicted hump-shaped relationships between precipitation and the abundance, richness, and biomass of macro-decomposers that peak in arid ecosystems.

## Results

Litter removal rate differed across seasons, sites, and mesh sizes, and all interactions between these factors were found significant as well (*Figure 3*, *Table 2*). On average, the litter removal rate was 2.6-fold higher in winter than in summer, 1.6-fold higher in meso- than in micro-baskets, and 1.3-fold higher in macro- than in meso-baskets. Litter removal was negligible in the hyper-arid sites during both seasons, while it was highest in the arid sites during summer and in the more mesic sites during winter

**Table 2.** Results of a full-factorial analysis of variance in litter removal rate across mesh sizes, experimental sites, and seasons.

| | Df | Sum Sq | Mean Sq | *F* value | Pr(>*F*) |
|---|---|---|---|---|---|
| Site | 6 | 0.002092 | 0.000349 | 171.946 | <0.001 |
| Season | 1 | 0.000679 | 0.000679 | 334.94 | <0.001 |
| Mesh size | 2 | 0.000267 | 0.000133 | 65.77 | <0.001 |
| Site:season | 6 | 0.00112 | 0.000187 | 92.044 | <0.001 |
| Site:mesh size | 12 | 0.000392 | 3.27E−05 | 16.11 | <0.001 |
| Season:mesh size | 2 | 1.77E−05 | 8.9E−06 | 4.368 | 0.0129 |
| Site:season:mesh size | 12 | 0.000129 | 1.08E−05 | 5.306 | <0.001 |
| Residuals | 1008 | 0.002044 | 0.000002 | | |

(*Figure 3*). Within site and season comparisons between mesh sizes yielded significant differences only in Avdat, Sayeret Shaked, and Havat Shikmim, indicating that faunal effects on decomposition were found only under arid to dry-semiarid conditions (*Figure 3*). Both macro- and mesofaunal effects were detected in the arid sites (Avdat and Sayeret Shaked), whereas the semiarid Havat Shikmim site exhibited only a mesofaunal effect during both seasons (*Figure 3*). The macrofaunal, mesofaunal, and microorganismal contributions to litter mass loss peaked under arid, semiarid and Mediterranean climate conditions, respectively (*Figure 4*). Whole-community litter removal rates were dictated by microorganisms in winter and by macrofauna in summer, resulting in comparable rates across the aridity gradient from Mediterranean to arid climate at the annual scale (*Figure 4*). In total, the whole-community litter removal rate peaked in Sayeret Shaked (MAP = 148 mm) and significantly decreased under drier and wetter conditions (*Figure 4*).

Macro-decomposer abundance, biomass, and morphospecies richness peaked in the arid sites during both seasons (*Figure 5*). The macro-decomposer assemblage differed significantly across sites ($F_{6,229}$ = 10.6, p-value <0.001) and across seasons ($F_{1,229}$ = 13.1, p-value <0.001), where woodlice, millipedes, and snails were substituted by ants and termites with increasing aridity (*Figure 6A, B*). Assemblage was significantly affected by the interaction between site and season too ($F_{6,229}$ = 5.4, p-value <0.001). The experimental site explained much of the assemblage variability across traps ($R^2$ = 0.18), whereas experimental season accounted for a smaller fraction ($R^2$ = 0.04), and site–season interaction played an intermediate role ($R^2$ = 0.10). All pairwise comparisons across sites yielded significant differences in assemblage (*Supplementary file 1*). In general, ants were the most abundant group, whereas beetles accounted for most of the biomass. However, under mesic conditions, woodlice (Ramat Hanadiv site), millipedes (Ramat Hanadiv and Bet Guvrin), and snails (Havat Shikmim) were dominant (*Figure 6C*). The Ramat Hanadiv assemblage was distinctively different from all other sites (*Figure 6A*), as demonstrated by very high Bray–Curtis (BC) dissimilarity indices compared to the other sites, regardless of the season (*Supplementary file 2*). There were parallels between the spatial and temporal axes of aridity, as winter communities in the most arid sites (Nahal Shita, Meishar, and Avdat) were relatively similar to the summer communities of the more mesic sites (Sayeret Shaked, Havat Shikmim, and Bet Guvrin) (*Figure 6A, B*; *Supplementary file 2*). The arid sites, where macro-decomposer assemblages flourished and were responsible for the highest litter mass loss, showed interesting seasonal dynamics. BC dissimilarity across seasons was higher in Sayeret Shaked than in Avdat (BC = 0.79 and 0.72, respectively). Cross-site dissimilarity between Sayeret Shaked and Avdat was higher in winter than in summer (BC = 0.85 and 0.72, respectively).

## Discussion

Our goal was to investigate how climate interacts with soil biota of different size categories to influence litter decomposition. We used litter baskets of different mesh sizes that were placed along a sharp precipitation gradient during hot–dry summer and again during colder–wetter winter. Our results suggest that decomposers respond differently to precipitation levels based on their size, leading to similar overall decomposition rates across the gradient, except in hyper-arid sites. We

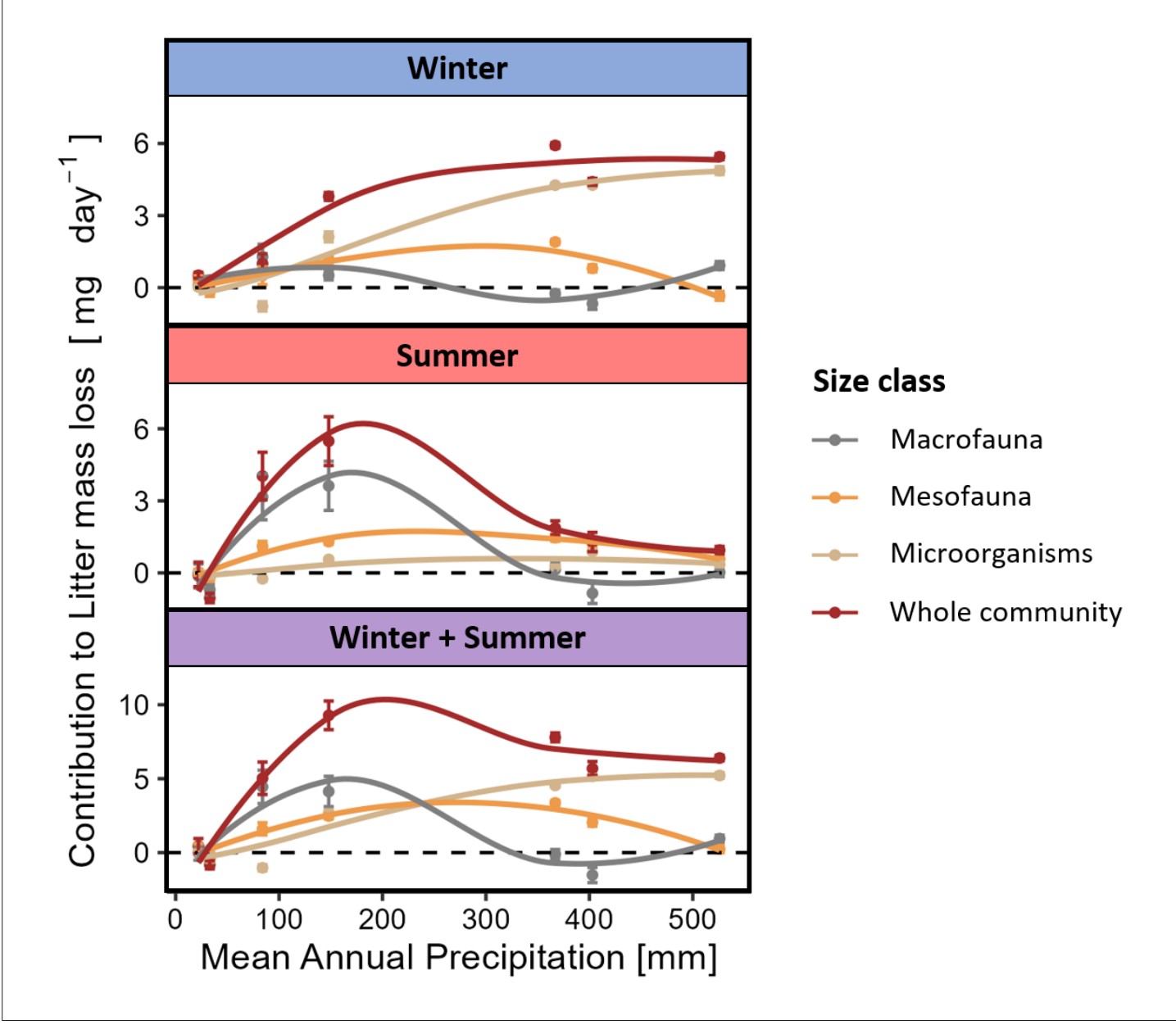

**Figure 4.** Contribution of different organism size classes to litter removal (mean ± standard error [se]) across the precipitation gradient during summer, winter, and both seasons combined. Macrofaunal contribution was calculated as the within-block difference between macro- and meso-baskets; mesofaunal contribution as the difference between meso- and micro-baskets; microbial and whole-community contributions represent litter removal rates in the micro- and macro-baskets, respectively. 25 litter baskets of each size class were used in each site during each season (total n = 1050). Curves were fitted to data using local estimation scatterplot smoothing (LOESS). Negative values may represent cases in which physical cleaning and ash correction failed to correct for all dust accumulation on the litter or cases in which exogenous litter may have penetrated the baskets. Our findings were not sensitive to these negative values.

found that microbial decomposition was minimal during the summer. In the winter, microbial decomposition was positively associated with precipitation, governing the whole-community decomposition. In both seasons, mesofaunal decomposition was moderate and followed a hump-shaped response to precipitation, peaking in semiarid sites. Macro-decomposition contributed minimally to whole-community decomposition during the winter, but during the summer dominated decomposition in the two arid sites. Using pitfall trapping, we found that macro-decomposer richness, abundance, and biomass followed a hump-shaped response to precipitation, peaking in arid environments.

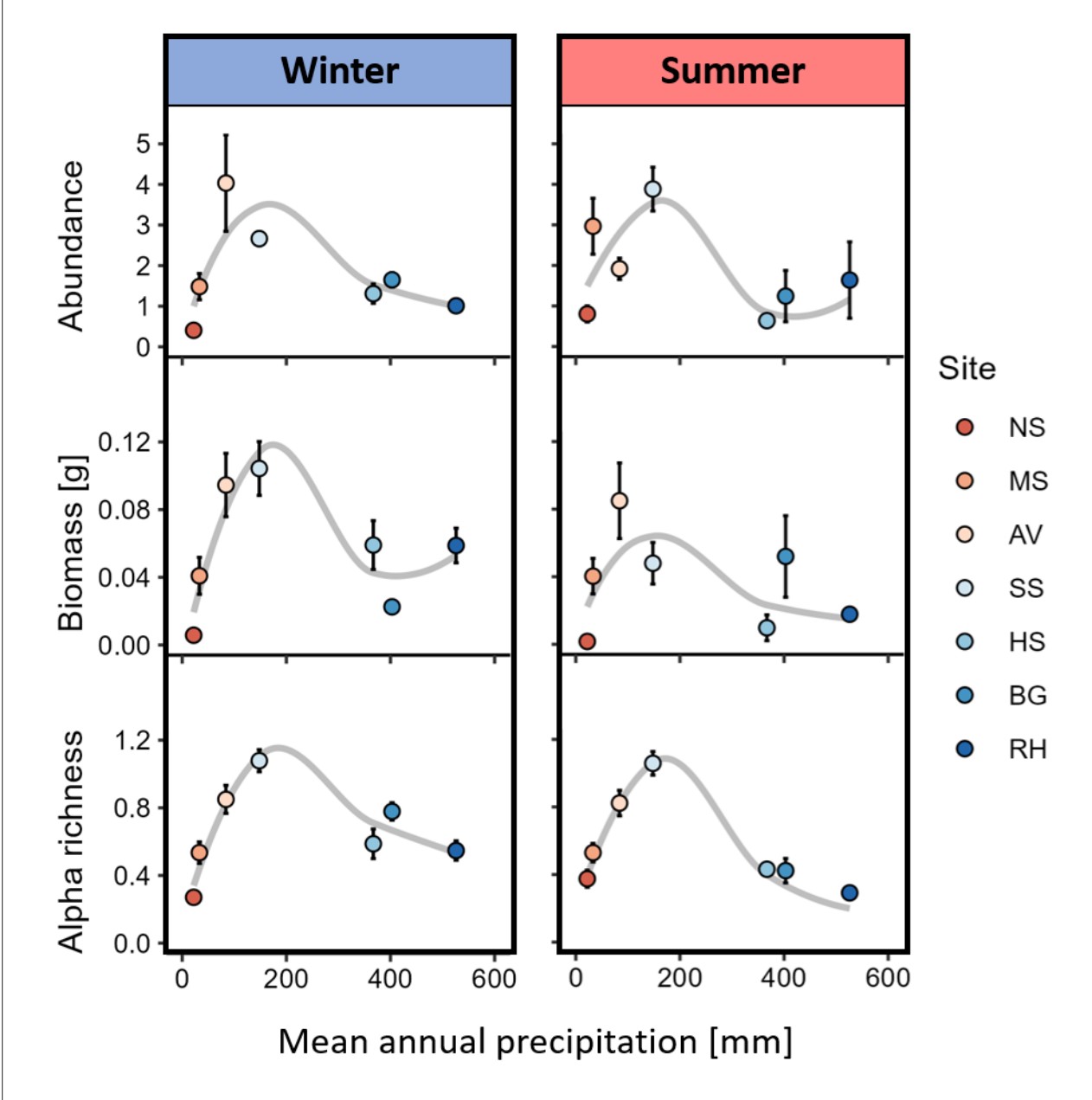

**Figure 5.** Macro-decomposer abundance, biomass, and alpha morphospecies richness across the precipitation gradient in the two experimental seasons (mean ± standard error [se]). Values are averaged across pitfall traps and divided by the number of trapping days. Sample sizes (number of recovered traps) in winter and summer, respectively: NS - 16,16; MS - 19,17; AV - 20,16; SS - 20,20; HS - 18,17; BG - 20,13; RH - 19,12. Curves are fitted to data using local estimation scatterplot smoothing (LOESS). NS – Nahal Shita; MS – Meishar; AV – Avdat; SS – Sayeret Shaked; HS – Havat Shikmim; BG – Bet Guvrin; RH – Ramat Hanadiv.

The puzzle of why plant litter decomposition in arid-lands is decoupled from annual precipitation and is faster than expected based on microbial decomposition models has bothered scientists for half a century (*Meentemeyer, 1978*; *Noy-Meir, 1974*; *Whitford et al., 1981*), and was later termed the dryland decomposition conundrum (*Throop and Archer, 2009*). Attempts to resolve this conundrum have focused predominantly on abiotic weathering agents, such as photodegradation (*Austin and Vivanco, 2006*; *Austin, 2011*) and thermal degradation (*Day et al., 2019*), alternative sources of moisture such as fog, dew, and atmospheric water vapor (*Evans et al., 2020*) and soil–litter mixing (*Throop and Archer, 2009*; *Throop and Belnap, 2019*). We, in turn, hypothesized that the opposing climatic dependencies of macrofauna and that of microorganisms and mesofauna should lead to

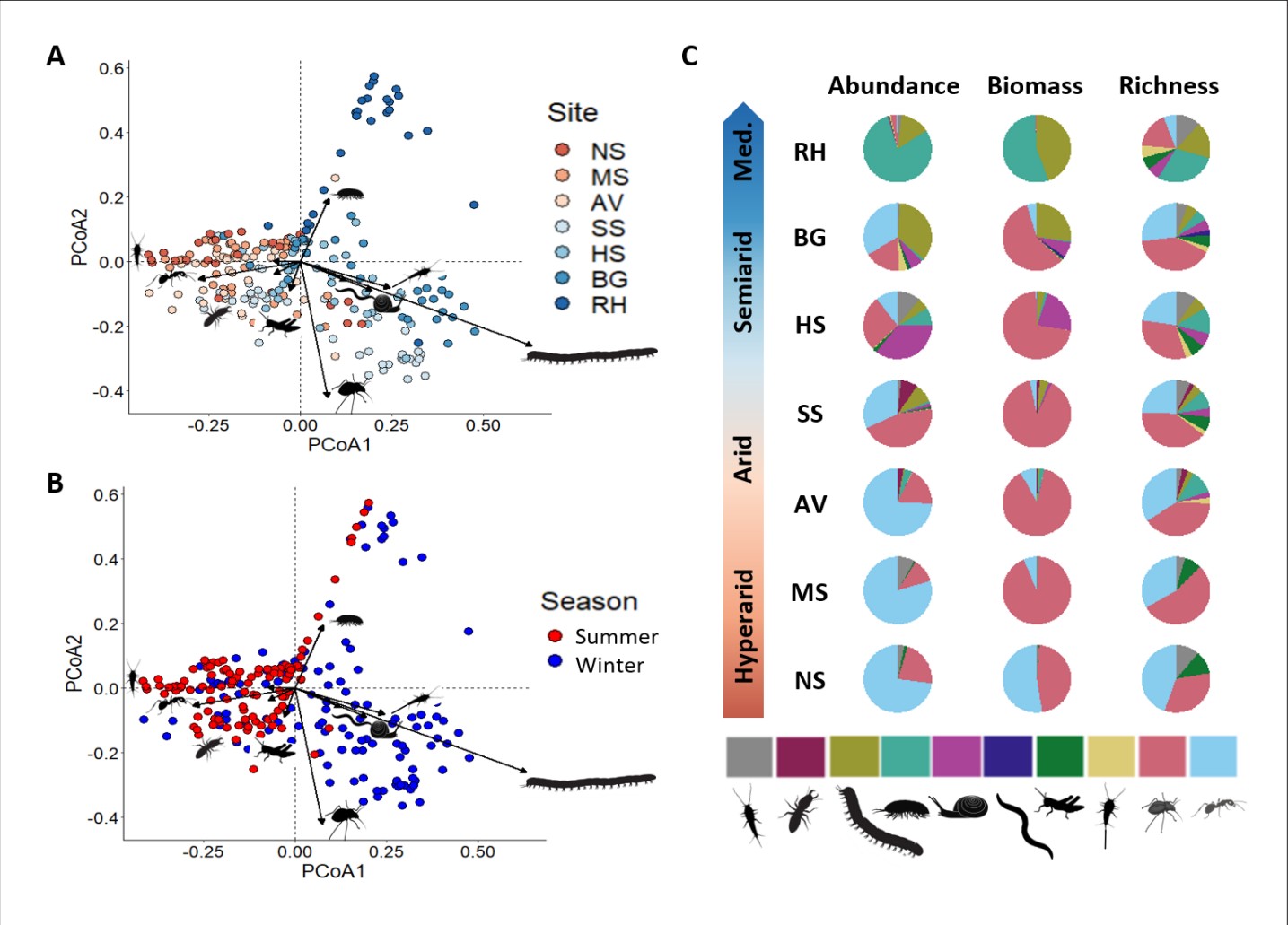

**Figure 6.** Differences in macro-decomposer assemblage across sites and seasons. (**A, B**) Graphical representation of the first two axes of a principal coordinate analysis on the macro-decomposer assemblage data. Colors represent experimental sites in A and seasons in B. Arrows represent taxonomic group scores fitted onto the principal coordinates analysis (PCoA) ordination. (**C**) Distribution of abundance, biomass, and morphospecies richness among macro-decomposer taxonomic groups in each site across the aridity gradient. NS – Nahal Shita; MS – Meishar; AV – Avdat; SS – Sayeret Shaked; HS – Havat Shikmim; BG – Bet Guvrin; RH – Ramat Hanadiv. Color codes (left to right in panel C): gray – Zygentoma, burgundy – termites, olive green – millipedes, turquoise – woodlice, pink – snails and slugs, purple – earthworms, dark green – crickets, pale yellow – bristletails, red – beetles, pale blue – ants.

similar overall decomposition rates across precipitation gradients, except in hyper-arid environments in which decomposers activity is predicted to be minimal regardless of organism size. Our results largely agree with this hypothesis. Whole-community decomposition was minimal in hyper-arid sites in both summer and winter. In the winter, microbial decomposition dominated the whole-community decomposition, demonstrating a positive response to precipitation that reached a maximum in the most mesic Mediterranean site. In contrast, macro-decomposition has contributed only little to the whole-community decomposition during the winter, but dominated the arid sites' decomposition in the summer. These findings supported the long-suggested but largely overlooked hypothesis that macro-decomposition governs plant litter decomposition in deserts (*Meentemeyer, 1978*; *Noy-Meir, 1974*; *Whitford et al., 1981*). The opposing climatic dependencies of micro- and macro-decomposers have led to similar or even higher annual decomposition rates in arid sites compared to those measured in more mesic sites. Consequently, we highlighted that differential climatic dependencies of different-sized decomposers rather than abiotic factors explain the discrepancy between classic decomposition models and the observed decomposition rates in drylands. This realization provided a plausible

resolution to the longstanding dryland decomposition conundrum, and exposed a hidden mechanism that may account for unexplained variation in plant litter decomposition across biomes.

Canonically, faunal decomposition is expected to be positively associated with temperature and moisture (*García-Palacios et al., 2013*). We, however, hypothesized that climate dependencies of mesofauna and macrofauna should differ due to the lower sensitivity of macrofauna to high temperature and low moisture, and the ability of macro-decomposers to shuttle between the hostile environment aboveground and the climatic havens belowground (*Sagi and Hawlena, 2021*). We also hypothesized that low and unpredictable resource availability in hyper-arid environments should limit macro-decomposer populations. Consequently, we predicted that macro-decomposers should be more prevalent in arid environments in comparison to hyper-arid or more mesic environments. Furthermore, ample resource availability may increase niche space (*Macarthur, 1965*), resulting in higher macro-decomposer diversity, which in turn can facilitate decomposition through synergistic effects of functionally complementary species (*Heemsbergen et al., 2004*; *Gessner et al., 2010*; *Boyero et al., 2021*; *Zeng et al., 2023*). Thus, we predicted that macro-decomposition should reflect the variation in the abundance, richness and biomass of macro-decomposers across the precipitation gradient. Our findings supported these predictions. The richness, abundance, and biomass of macro-decomposers followed a hump-shaped relationship with precipitation, peaking in arid environments and diminishing toward hyper-arid or semiarid and Mediterranean sites. Macro-decomposer assemblages were dominated by ants and beetles across the aridity gradient except in the Mediterranean site that was dominated by isopods and millipedes. During the summer, the observed hump-shaped relationship between macro-decomposition and precipitation tightly echoed the variation in richness, abundance, and biomass of macro-decomposers, revealing the mechanistic foundation for the cross-system variation in macro-decomposition.

In winter, macro-decomposer abundance, richness, and biomass were similar to or even higher than those measured during the summer across all sites. Despite this resemblance, macro-decomposition did not reflect the observed variation in macro-decomposer assemblages. This discrepancy could be explained by between-seasons differences in the structure of the macro-decomposer assemblages (*Figure 6B*). The macro-decomposer summer assemblage in Sayeret Shaked was more similar to the Avdat summer assemblage than to the Sayeret Shaked winter assemblage. The Avdat assemblages were more similar to each other across seasons than the Sayeret Shaked assemblages. This may explain why macro-decomposition in winter was higher in Avdat than in Sayeret Shaked. Termites (*Hodotermitidae* sp.), that play an important role in decomposition, were more abundant in summer compared to the winter in Sayeret Shaked but not in Avdat. Moreover, our data revealed that several beetle taxa (*Adelostoma* sp., *Akis reflexa* (Fabricius, 1775), *Dailognatha crenata* (Reiche & Saulcy, 1857), *Tentyrina* sp., and *Zophosis* sp.) were prevalent in both arid sites during the summer but were absent or very scarce in the winter. Phenological differences in the behavior of dominant macro-decomposers may also contribute to the seasonal differences (*Bonato Asato et al., 2023*). For instance, *Hemilepistus reaumuri* (Milne Edwards, 1840), an abundant isopod species in Avdat, and *Anacanthotermes ubachi* (Navás, 1911), a common termite species in Sayeret Shaked, consume detritus predominantly during the summer and autumn and disperse and reproduce during the winter (*Zaady et al., 2003*; *Shachak et al., 1976*). Future studies should explicitly test these explanations. Regardless, the whole-year association between macro-decomposition and the abundance, richness, and biomass of macro-decomposers strongly support our hypothesis.

Theory suggests that plant litter decomposition by meso-decomposers should increase with moisture. This pattern was supported by a cross-biome experiment (*Wall et al., 2008*). Thus, we hypothesized that both meso-decomposition and microbial decomposition should increase with precipitation and be more prominent in the winter than in the summer. Our results did not coincide with these hypotheses. Litter decomposition by mesofauna followed a unimodal pattern across the precipitation gradient, peaking under semiarid conditions in both seasons (*Figure 4*). Meso- and macro-decomposition were similar in the hyper-arid and Mediterranean sites. However, meso-decomposition was higher than macro-decomposition in the semiarid sites and much lower than macro-decomposition in the arid sites. These results suggest that meso-decomposers, like macro-decomposers, have adaptations that allow them to strive in moisture-deprived environments. Yet, meso-decomposition peaked in more mesic conditions than macro-decomposition, implying higher moisture dependency. Our findings highlight the need to explore the moisture dependency of mesofauna in greater details,

and to generate different predictions for how mesofauna and microorganisms may affect plant litter decomposition.

Faunal decomposition in our study peaked in arid environments, contrasting the positive association between faunal decomposition and precipitation that was found in recent global meta-analyses (*García-Palacios et al., 2013*; *Xu et al., 2020*). This discrepancy may reflect underestimation of faunal decomposition rates in drylands, possibly because these studies either deliberately grouped cold and dry environments together (*García-Palacios et al., 2013*; *Wall et al., 2008*), or focused solely on precipitation without accounting for differences in temperature (*Xu et al., 2020*). In cold water-deprived environments and seasons, low temperatures may limit the populations and activity of ectotherm animals (*Swift et al., 1979*; *Johnston and Sibly, 2020*). Therefore, ignoring the effect of temperature may lead to falsely smaller faunal effects on decomposition in drylands. This bias may contribute to the positive association between precipitation and faunal decomposition. To reveal the realistic relationships, future studies on faunal decomposition should explore the effects of temperature, precipitation, and the interaction between them. It is important noting that temperature could affect decomposition both directly by determining the activity of ectotherms and indirectly by regulating moisture availability (*Sagi and Hawlena, 2024*). Thus, using aridity indices that aim to correct for moisture availability cannot resolve the need to account also for temperature per se.

We used litter of one plant species that is common to all seven experimental sites. In this way, we were able to compare decomposition rates across sites without using litter that is exogenous to one or more sites. Decomposer communities are assumed to perceive litter quality based on the chemical composition of the focal litter, and that of past resource inputs that these communities have experienced (*Strickland et al., 2009*). Plant communities differ substantially between our sites, suggesting that each decomposer community has experienced very different resource inputs, and may have access to alternative litter of various qualities. Thus, it is likely that the perceived quality of *S. capensis* litter differs between sites, possibly confounding our results. Moreover, our sites were distributed along a single precipitation gradient. Thus, our results may be confounded by other environmental factors that were not accounted for. To determine how general our findings are, future research should repeat our experimental approach using different litter sources, and across multiple climatic gradients.

In conclusion, our work revealed that decomposers of varying size categories have different moisture dependencies. This suggests that microorganisms, meso-decomposers, and macro-decomposers should be considered separately in decomposition models, and emphasizes the need to contemplate animals' physiology and behavior when investigating zoogeochemical processes. Warm drylands cover 19% of the land surface worldwide and expand rapidly due to unsustainable land-use and climate change (*Cartereau et al., 2022*). We highlight the importance of macro-decomposition in arid-lands that compensates for the minimal microbial decomposition, providing a plausible resolution to the long-debated dryland decomposition conundrum. Understanding the mechanisms that regulate decomposition in drylands is key for conserving and restoring fundamental ecosystem processes in these ever-growing areas, and in improving our understanding of global processes like C cycling. To date, the general conceptualization of decomposition is largely based on ample research from temperate ecosystems. Thus, prevailing theory centers on focal processes that dominate decomposition in these systems. Our work highlights that in other less studied ecosystems additional processes like the role of animal decomposers may be dominant, opening the door for new exciting research that may shake our conceptualization of decomposition processes.

## Materials and methods

We performed a manipulative litter mass loss experiment across seven sites representing a sharp MAP gradient ranging from hyper-arid desert to Mediterranean maquis (*Figure 2A*, *Table 1*). All sites were chosen to be on calcareous soils formed upon sedimentary limestone rock in natural habitats. The mean annual temperature varies only slightly across sites from 18.7 to 22.3°C. The exact study sites were determined to ensure minimal human disturbance during the year-long experiment. In each of the seven sites we installed litter baskets of three different mesh sizes that control organismal access to litter: micro-baskets allowing entry of only microorganisms (<200 µm), meso-baskets allowing entry of microorganisms and mesofauna (<2 mm), and macro-baskets that were identical to the meso-baskets but with side openings that allow entry of macrofauna (<2 cm). Litter baskets were filled with

leaf litter belonging to the annual grass *Stipa capensis* Thunb. that is native to all seven study sites. This approach allowed us to compare decomposition rates across sites without using exogenous litter that may decompose in a very different rate than local litter (*Joly et al., 2023*). Twenty-five blocks, each including the three basket types (*Figure 2B*), were installed in each site for two consecutive experimental periods – a wet cool winter and a dry hot summer (2 periods × 7 sites × 3 treatments × 25 blocks = 1050 baskets in total). We also characterized the macro-decomposer assemblage in each site during the two seasons using pitfall trapping.

## Litter basket experiment

We collected *S. capensis* litter from the Avdat site in the summer of 2020 and air dried it. We sorted the litter to remove litter belonging to any other species and assigned 3 ± 0.0001 g (Mettler Toledo MS105DU) to each litter basket. Thirty additional litter samples were oven dried at 60°C for 48 hr and weighed again for determination of initial moisture content. The 14 × 13 × 3.6 cm litter baskets were prepared of a 12-mm mesh galvanized welded metal, lined at the bottom with a 1.5-mm fiberglass mesh to prevent litter loss, and covered from all sides (including top and bottom) with a 2-mm metal mesh to exclude termites. In the macro-baskets, three 2 × 2 cm windows were cut at each of the four sides. The windows were cut approximately 1 cm above ground level to allow macrofaunal access but prevent accidental litter spill. This may slightly reduce macrofaunal access, making our estimations of the macrofaunal effect conservative. In the micro-baskets, we placed the litter within a polyethylene 200-µm mesh bag. In the macro- and meso-baskets, we laid a 2-cm heavy metal mesh over the litter to minimize litter loss due to wind.

We installed the first batch of 525 litter baskets in the field in November–December 2020. All blocks were placed around similarly sized bushes of locally distributed species and tethered to the ground using metal stakes. We collected the baskets in May–July 2021 and replaced them with a new similar batch that was later collected in October–November 2021. At the end of each season, the collected baskets were transported to the laboratory in sealed Ziplock bags. Any litter spilled during transportation was weighed and the weight loss was incorporated in the calculations. Leaf litter in each basket was first screened for adulteration from leaf litter of other species, following which the *S. capensis* litter was oven dried at 60°C for 48 hr and weighed. To account for dust accumulation on the litter we applied an ash correction procedure (*Barney et al., 2015*). We burned and weighed five sub-samples from each site–season–treatment combination (550°C for 5 hr) and calculated the combination-specific mean ash content. The final litter mass was corrected for ash content based on these calculations. We burned and weighed 15 additional samples of *S. capensis* litter that were not placed in the field and calculated the mean ash content of the initial litter. The initial litter mass was corrected accordingly. The rate of litter removal from each basket was calculated as the difference between the ash corrected final dry litter mass and the ash and moisture corrected initial litter mass, divided by the number of days the litter spent in the field.

## Pitfall trapping

We characterized the macro-decomposer assemblages by setting up 20 pitfall traps for 5–7 days at each site during each experimental period. Wet season traps were opened in February 2021, whereas the dry season traps were opened between late August and early October. We installed traps by placing two 10 cm diameter × 7.5 cm deep plastic containers one inside another such that the opening was flushed with the ground. We added to each trap 150 ml of preservative, which comprised of 40% absolute ethanol, 40% distilled water, and 20% propylene glycol. Traps were covered with steel mesh of large mesh size to prevent small mammals and reptiles from falling inside. At the end of the 5–7 days, samples were collected and transferred to 70% ethanol. Samples were sorted and identified to morphospecies level in the lab. Only animals larger than 2 mm were included in the analysis. Sub-samples were freeze-dried and weighed (Mettler Toledo MS105DU) for biomass estimation of each morphospecies.

## Analytical procedures

We first fitted a linear mixed model to the litter removal rate data, including experimental site, experimental season, mesh size, and all interaction terms as fixed effects. The random effect of the experimental spatial blocks was found insignificant using a simulation test with 9999 simulations (RLRT =

2.1, p-value = 0.07). Therefore, we assessed the effects of the site, season, and mesh size on the litter removal rate using a full factorial analysis of variance, followed by Tukey's Honestly Significant Difference pairwise comparisons. We calculated the contribution of each size group to litter mass loss by block. Microbial contribution was defined as the mass loss from micro-baskets; mesofaunal contribution was calculated as the difference between mass loss from meso- and micro-baskets; macrofaunal contribution was calculated as the difference between mass loss from macro- and meso-baskets; whole-community decomposition was defined as the mass loss from macro-baskets. We modeled the relationship between MAP and each of these contributions using Locally Estimated Scatterplot Smoothing. We used the same method to model the relationship between MAP and the macro-decomposer abundance, biomass, and morphospecies richness in each season. We assessed differences in the macro-decomposer assemblage among experimental sites and seasons using a principal coordinates analysis (PCoA) with individual traps as the sampling units and BC index as the dissimilarity metric. We tested for differences across sites and seasons in macro-decomposer assemblage using a permutational multivariate analysis of variance, followed by pairwise comparisons between sites using the Benjamini–Hochberg p-value adjustment. BC indices between site–season combinations were calculated as well, based on the summed abundances across traps. To explore which macro-decomposer groups dominate the different sites and seasons, we classified the identified morphospecies to ten macro-decomposer taxa: Archaeognatha (bristletails), Coleoptera (beetles), Diplopoda (millipedes), Formicidae (ants), Gastropoda (snails and slugs), Grylloidea (crickets), Isoptera (termites), Lumbricina (earthworms), Oniscidea (woodlice), and Zygentoma. Then we summed the abundance, richness, and biomass from each group in each trap. We used the abundance data to fit the group scores onto the PCoA ordination. Litter removal data was analyzed using the 'stats', 'lme4' (*Bates et al., 2015*), and 'RLRsim' (*Scheipl et al., 2008*) packages from R software (version 4.3.0) (*R Development Core Team, 2022*), whereas assemblage data was analyzed using the 'vegan' package (*Oksanen et al., 2022*).

## Acknowledgements

We thank Liat Hadar, Ronen Kadmon, Ronen Ron, Omri Sharon, Shayli Dor-Haim, Nitzan Segev, and Rachel Armoza-Zvuloni for facilitating our work in the experimental sites; Igor Armiach Steinpress, Laibale Friedman, and Armin Ionescu for help with morphospecies identification; and Ariel Malinsky-Buller, Netanel Paz, Omri Sherman, Dor Gabay, Coral Ben-Lulu, Ofer Frumkin, Aparna Lajmi, and Atar, Alon and Ela Hawlena for lab and field assistance.

## Additional information

### Funding

| Funder | Grant reference number | Author |
|---|---|---|
| Israel Science Foundation | ISF-No.1391/19 | Dror Hawlena |

The funders had no role in study design, data collection and interpretation, or the decision to submit the work for publication.

### Author contributions

Viraj R Torsekar, Conceptualization, Data curation, Investigation, Methodology, Writing – original draft, Writing – review and editing; Nevo Sagi, Conceptualization, Formal analysis, Investigation, Methodology, Writing – original draft, Writing – review and editing; J Alfred Daniel, Data curation, Methodology, Writing – review and editing, Identified animal samples; Yael Hawlena, Data curation, Methodology, Writing – review and editing, Identified animal samples; Efrat Gavish-Regev, Data curation, Methodology, Writing – review and editing, Identified animal samples; Dror Hawlena, Conceptualization, Supervision, Funding acquisition, Investigation, Methodology, Writing – original draft, Project administration, Writing – review and editing

### Author ORCIDs

Viraj R Torsekar ⬤ https://orcid.org/0000-0001-6096-6454

Nevo Sagi http://orcid.org/0000-0002-4745-4304
Dror Hawlena http://orcid.org/0000-0001-9142-1553

Reviewer #1 (Public review): https://doi.org/10.7554/eLife.93656.3.sa1
Author response https://doi.org/10.7554/eLife.93656.3.sa2

## Additional files

### Supplementary files
• Supplementary file 1. Results of post hoc pairwise comparisons between macro-decomposer assemblages across experimental sites.

• Supplementary file 2. Dissimilarity matrix between macro-decomposer assemblages of the different site–season combinations.

• MDAR checklist

### Data availability
All data used in the manuscript has been deposited in an open repository and can be found on the following link https://doi.org/10.6084/m9.figshare.23544993.

The following dataset was generated:

| Author(s) | Year | Dataset title | Dataset URL | Database and Identifier |
|---|---|---|---|---|
| Torsekar VR, Sagi N, Alfred Daniel J, Hawlena Y, Gavish-Regev E, Hawlena D, Alfred Daniel J | 2024 | Contrasting responses to aridity by different-sized decomposers cause similar decomposition rates across a precipitation gradient | https://doi.org/10.6084/m9.figshare.23544993 | figshare, 10.6084/m9.figshare.23544993 |

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
