## [Editor Report · eLife Assessment]

This **fundamental** study substantially advances our understanding of the role of different-sized soil invertebrates in shaping the rates of leaf litter decomposition, using an experiment across seasons along an aridity gradient. The authors provide **compelling** evidence that the summed effects of all invertebrates (with large-sized invertebrates being more active in summer and small-sized invertebrates in winter) on decomposition rates result in similar levels of leaf litter decomposition across seasons. The work will be of broad relevance to ecosystem ecologists interested in soil food webs, and researchers interested in modeling carbon cycles to understand global warming.

---

## [Referee Report · Reviewer #1 (Public review)]

Summary:

Torsekar et al. use a leaf litter decomposition experiment across seasons, and in an aridity gradient, to provide a careful test of the role of different-sized soil invertebrates in shaping the rates of leaf litter decomposition. The authors found that large-sized invertebrates are more active in the summer and small-sized invertebrates in the winter. The summed effects of all invets then translated into similar levels of decomposition across seasons. The system breaks down in hyper-arid sites.

---

## [Author Response]

The following is the authors’ response to the original reviews.

**Public Reviews:**

**Reviewer #1 (Public Review):**
Summary:I really enjoyed this manuscript from Torsekar et al on "Contrasting responses to aridity bydifferent-sized decomposers cause similar decomposition rates across a precipitation gradient". The authors aimed to examine how climate interacts with decomposers of different size categories to influence litter decomposition. They proposed a new hypothesis: "The opposing climatic dependencies of macrofauna and that of microorganisms and mesofauna should lead to similar overall decomposition rates across precipitation gradients".This study emphasizes the importance as well as the contribution of different groups of organisms (micro, meso, macro, and whole community) across different seasons (summer with the following characteristics: hot with no precipitation, and winter with the following characteristics: cooler and wetter winter) along a precipitation gradient. The authors made use of 1050 litter baskets with different mesh sizes to capture decomposers contribution. They proposed a new hypothesis that was aiming to understand the "dryland decomposition conundrum". They combined their decomposition experiment with the sampling of decomposers by using pittfall traps across both experiment seasons. This study was carried out in Israel and based on a single litter species that is native to all seven sites. The authors found that microorganism contribution dominated in winter while macrofauna decomposition dominated the overall decomposition in summer. These seasonality differences combined with the differences in different decomposers groups fluctuation along precipitation resulted in similar overall decomposition rates across sites.I believe this manuscript has a potential to advance our knowledge on litter decomposition.Strengths:Well design study with combination of different approaches (methods) and consideration of seasonality to generalize pattern.The study expands to current understanding of litter decomposition and interaction between factors affecting the process (here climate and decomposers).Weaknesses:The study was only based on a single litter species.

We now discuss the advantages and limitations of this approach in the methods and devote a completely new paragraph to this important point in the discussion (lines 394-401).

**Reviewer #2 (Public Review):**
Summary: Torsekar et al. use a leaf litter decomposition experiment across seasons, and in an aridity gradient, to provide a careful test of the role of different-sized soil invertebrates in shaping the rates of leaf litter decomposition. The authors found that large-sized invertebrates are more active in the summer and small-sized invertebrates in the winter. The summed effects of all invets then translated into similar levels of decomposition across seasons. The system breaks down in hyper-arid sites.Strengths: This is a well-written manuscript that provides a complete statistical analysis of a nice dataset. The authors provide a complete discussion of their results in the current literature.Weaknesses:I have only three minor comments. Please standardize the color across ALL figures (use the same color always for the same thing, and be friendly to color-blind people).

Thank you for this important suggestion. We have now changed all figures to standardize all colors and chose a more color-blind friendly pallete.

Fig 1 may benefit from separating the orange line (micro and meso) into two lines that reflect your experimental setup and results. I would mention the dryland decomposition conundrum earlier in the Introduction.

We based our novel hypotheses on a thorough literature search. Accordingly, decomposition is expected to be positively associated with moisture, regardless of the decomposer body size. Our contribution to theory was to suggest that macro-detritivores may respond very differently to climatic conditions and dominate litter decomposition in warm arid-lands (we listed the reasons in the text). Consequently, we did not distinguish between microorganisms and mesofauna. We assumed that both groups inhabit the litter substrate and have limited adaptation to dry conditions. Our results provide strong evidence that this presumption is likely wrong and that mesofauna respond to climate very differently from micro-decomposers. Yet, we cannot use hindsight understanding to improve our original hypothesis. We now emphasize this important point at the discussion as important future direction.

Although we are very appreciative and pleased with the reviewer enthusiasm to highlight the importance of our work as a possible solution to the longstanding dryland decomposition conundrum, we decided not to move it to the introduction. This is because we think that our work is not centred on resolving the DDC but provides more general principles that may lead to a paradigm shift in the way ecologists study nutrient cycling across ecosystems.

And the manuscript is full of minor grammatical errors. Some careful reading and fixing of all these minor mistakes here and there would be needed.

We apologize and did our best to find and fix those mistakes

**Recommendations for the authors:**

**Reviewer #1 (Recommendations For The Authors):**
I really enjoyed this manuscript from Torsekar et al on "Contrasting responses to aridity by different-sized decomposers cause similar decomposition rates across a precipitation gradient". The authors aimed to examine how climate interacts with decomposers of different size categories to influence litter decomposition. They proposed a new hypothesis: "The opposing climatic dependencies of macrofauna and that of microorganisms and mesofauna should lead to similar overall decomposition rates across precipitation gradients".This study emphasizes the importance as well as the contribution of different groups of organisms (micro, meso, macro, and whole community) across different seasons (summer with the following characteristics: hot with no precipitation, and winter with the following characteristics: cooler and wetter winter) along a precipitation gradient. The authors made use of 1050 litter baskets with different mesh sizes to capture decomposers contribution. They proposed a new hypothesis that was aiming to understand the "dryland decomposition conundrum". They combined their decomposition experiment with the sampling of decomposers by using pitfall traps across both experiment seasons. This study was carried out in Israel and based on a single litter species that is native to all seven sites. The authors found that microorganism contribution dominated in winter while macrofauna decomposition dominated the overall decomposition in summer. These seasonality differences combined with the differences in different decomposers groups fluctuation along precipitation resulted in similar overall decomposition rates across sites.I believe this manuscript has the potential to advance our knowledge on litter decomposition. Below i provide my general and specific comments.General comments:(1) Study in general is well designed and well thought beforehand,(2) Study aims to expand the current understanding of the dryland decomposition conundrum(3) The should put a caveat to the fact they only use one litter species and call for examining litter mixture in the same gradient.(4) Please check the way you reduce the random effects from your initial model, I have provided a better way to do so in my specific comments(5) For Figure 1, authors can check my comment on this and see if they could revise the figure.

Thank you for the positive feedback and your valuable comments. We have tried to best address all comments and suggestions for improvement and clarification

Specific commentsLine # 57 Please write "Theory suggests" instead of "Theory suggest"

We changed the text as suggested

Line # 70, please write "Indeed, handful evidence shows" instead of "Indeed, handful evidence show"

We changed the text as suggested

Figure 1: I like this conceptual framework. I have a silly question, why is it that the slopes of the whole community at the beginning (between Hyperarid and Arid) is the same as the Macro fauna, I would think the slope should be higher as this is adding up right? and also the same goes for the decomposition of whole community later on. For me this should reflect the adding or summing up (if i am right) then the authors should think about how this could be reflected in the figure.

We agree with your interpretation that the whole community decomposition reflects the addition by constituent decomposers. The slope of the whole community decomposition between hyper-arid and arid is slightly higher than the one of macro decomposition to reflect the additive effect of macro with meso+micro decomposition. We have now changed the figure slightly to make this point more visible (Line 106).

Line # 111 Please make "Methods" bold as well to be consistent with others headings.

We changed the formatting as suggested

Line #125 and in other lines as well please replace "X" by "x" to denote multiplication.

We changed the formatting as suggested

Table 1 Please add "*" to climate like this "Climate*" so that the end note of the table could make sense

Thank you for this suggestion. We have now added the asterisk referring to the note below the Table.

Figure 2, please consider putting at line #133, mean annual precipitation (MAP), as such for line # 135 You can directly says The precipitation map ....

We made both changes as suggested.

Line # 138 I would not use the different units for the same values. I do understand that you want to emphasize the accuracy but i would write instead 3 +- 0.001 g

We changed the units as suggested.

Line # 145, how is the litter basket customized to rest at 1 cm above ground level?

We have now clarified –that we cut-open windows one centimeter above the cage floor. The cages were positioned on the soil (line 144).

Lines # 181-183, I like the approach of checking the necessity of having the random effects. However, it has been reported that likelihood ratio test (LRT) are not really reliable to test for random effects. I will suggest you rather use permutations instead. I think the function is confint(MODEL) you need to specify the number of permutation the higher the better but you should start with 99 first and see how the results look like if promising then you can even go to 9999. But it will need computation power and time.

Thank you for the suggestion. We now used a simulation-based exact test, instead of a LRT, to examine the random effect, as recommended by the authors from the “lme4” package. As recommended, we used 9999 simulations. The simulation test yielded a similar result to those originally reported (see lines 181-183).

Line # 187, 188, 188, please do not use capital letter to start mesofauna, macrofauna and whole-community

We changed the formatting as suggested

Line # 205 Please add the version number of R in the text.

We now included the version number as suggested.

Line # 209-211, could you please check whether "then" is the word you want to use or "than"

Our bad- we indeed meant “than” and have made the appropriate changes.

Line # 227 and in other places as well please provide the second degree of freedom of the F test.

Thank you for this important comment. We have now added the second degree of freedom to the relevant results (lines 229, 232).

Figure 3 and Figure 4 show some results that are negative, can you please explain what might be the reasons behind this?

We now explain this important point in the figures’ captions.

Figure 5 Please add label to the x-axis.

Thank you-we have now included a label.

Line # 357, the sentence "... meso-decomposition, like microbial decomposition,...", I don't understand which criteria authors used to classify microbial decomposition as "meso-decomposition"?

We now remove this potential cause of confusion by using the term ‘meso-decomposition’ to distinguish from microbial decomposition (Line 366).

Line # 380 Kindly put "per se" in italic.

We changed the formatting as suggested

ReferencesThe references format are not consistent. For example for the same journal (say Trends in Ecology and Evolution) the authors sometimes wrote the full name like at line # 36 (and also realize that "vol" should not be written as such) but wrote the abbreviations at line #42

Our bad- we apologize and carefully checked all references to make sure the style is consistent.